# Foraging as sampling without replacement: A Bayesian statistical model for estimating biases in target selection

**Alasdair D. F. Clarke**[1], **Amelia R. Hunt**[2], **Anna E. Hughes**[1] *

**1** University of Essex, Department of Psychology, Colchester, United Kingdom, **2** University of Aberdeen, School of Psychology, Aberdeen, United Kingdom

* anna.hughes@essex.ac.uk

**Data Availability Statement:** Data and code are available at https://osf.io/7yuaz/.

**Funding:** This study was funded by the Economic and Social Research Council, Grant no: ES/

## Abstract

Foraging entails finding multiple targets sequentially. In humans and other animals, a key observation has been a tendency to forage in 'runs' of the same target type. This tendency is context-sensitive, and in humans, it is strongest when the targets are difficult to distinguish from the distractors. Many important questions have yet to be addressed about this and other tendencies in human foraging, and a key limitation is a lack of precise measures of foraging behaviour. The standard measures tend to be run statistics, such as the maximum run length and the number of runs. But these measures are not only interdependent, they are also constrained by the number and distribution of targets, making it difficult to make inferences about the effects of these aspects of the environment on foraging. Moreover, run statistics are underspecified about the underlying cognitive processes determining foraging behaviour. We present an alternative approach: modelling foraging as a procedure of generative sampling without replacement, implemented in a Bayesian multilevel model. This allows us to break behaviour down into a number of biases that influence target selection, such as the proximity of targets and a bias for selecting targets in runs, in a way that is not dependent on the number of targets present. Our method thereby facilitates direct comparison of specific foraging tendencies between search environments that differ in theoretically important dimensions. We demonstrate the use of our model with simulation examples and re-analysis of existing data. We believe our model will provide deeper insights into visual foraging and provide a foundation for further modelling work in this area.

## Author summary

Foraging has been well-studied in many species that rely on widely distributed food sources, such as bees and birds. Less well understood is how humans approach foraging tasks, and whether there are general policies we can identify that describe how we search for different categories of objects that can vary in quantity and distribution. We present a way to model foraging behaviour as a generative sampling without replacement procedure, implemented in a Bayesian multilevel model. This allows us to break down behaviour into a number of independent biases that influence target selection, including the

S016120/1 to AC and ARH. (https://esrc.ukri.org)
The funders had no role in study design, data
collection and analysis, decision to publish, or
preparation of the manuscript.

**Competing interests:** The authors have declared
that no competing interests exist.

proximity of targets, a bias for selecting targets in runs and a bias for a particular target type, in a way that is not dependent on the number of targets present. We believe this tool can open the door for foraging to become a standard task for refining our understanding of attention, working memory, prospective memory, learning, planning and decision-making.

## Introduction

Foraging typically refers to the process of searching for and gathering dispersed food. As would be expected given the survival value of this skill, a well-studied question in animal behaviour is the extent to which foraging can be thought of as efficient (e.g. [1, 2]). That is, does the forager's path and food source selection maximize energy intake and minimize energy expenditure and risk? For contemporary humans, most foraging for food takes place in supermarkets, but foraging can be more broadly defined as search for multiple instances of different categories of targets. Many tasks involve this more general definition of foraging (e.g. a security officer monitoring a crowded event; a radiologist searching for all signs of cancer in an X-ray [3]).

The human foraging literature has been directly inspired by animal behaviour research, including research on how humans find patches [4] and how they terminate their search and move onto another patch [3, 5–8]. In the latter context, an influential modelling framework has been marginal value theorem (MVT) [9], which predicts that an optimal forager should leave a patch when the "instantaneous rate of return" from a given patch drops below the average rate of return across all the patches (including travel time, during which no elements can be collected). Recent work has shown that human observers can forage optimally [3], although interestingly this may not hold in all cases e.g. older adults appear to show non-optimal behaviour, staying too long in each patch to adhere to MVT [8], and other studies have also found deviations from optimality [10]. Thus, when considering search termination, there are good theoretical models of behaviour that can be used to understand the cognitive processes underlying performance.

Another well-established feature of foraging is that animals tend to search for food in 'runs' of one particular food type, particularly when prey are cryptic [11, 12]. (See Figs 1 and 2 for examples.) Pollinators exhibit a tendency known as 'flower constancy', where they selectively visit flowers of a particular category and ignore equivalently rewarding flowers from other categories. This tendency seems to be context-sensitive in many species, suggesting it is a behavioural adaptation e.g. [13]. These observations led to research into the concept of the 'search image', where animals direct selective attention to particular features of prey in order to facilitate fast detection [14, 15]. A similar concept of 'attentional templates' (the representation of the search goal in working memory) has also been proposed for human visual search [16]. Target switching is important because it presents a promising model task for understanding the coordination of cognitive and physical effort. Selecting targets based on proximity, that is, selecting all the nearby targets before moving to a new area, minimizes the distance of travel, but the cost is the high cognitive load associated with switching between multiple target templates in working memory. Selecting all the targets of one category before moving on to the next category will minimize the cognitive load, at the cost of increased distance of travel between targets. Which of these two strategies is more efficient will depend on features of both the environment (the distance between targets, the number and complexity of the different types of targets) and the individual (their physical as well as cognitive capabilities). Whether

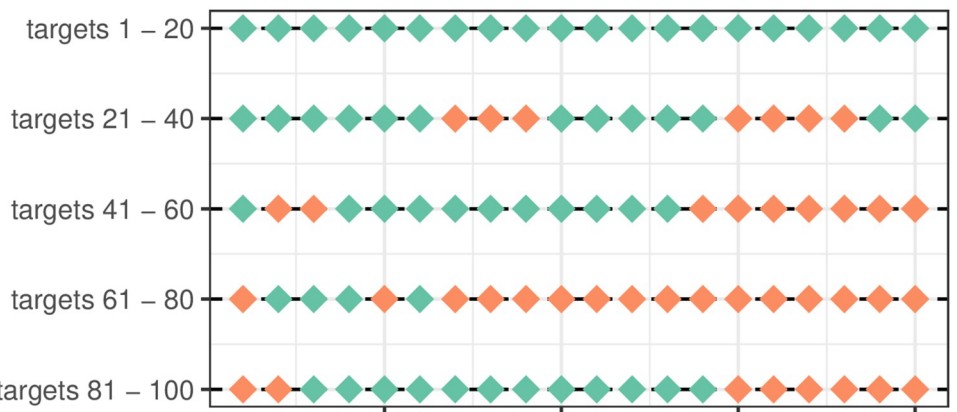

**Fig 1. Example data from Dawkins (1971).** The first 100 grains taken by one chick in experiment 1.

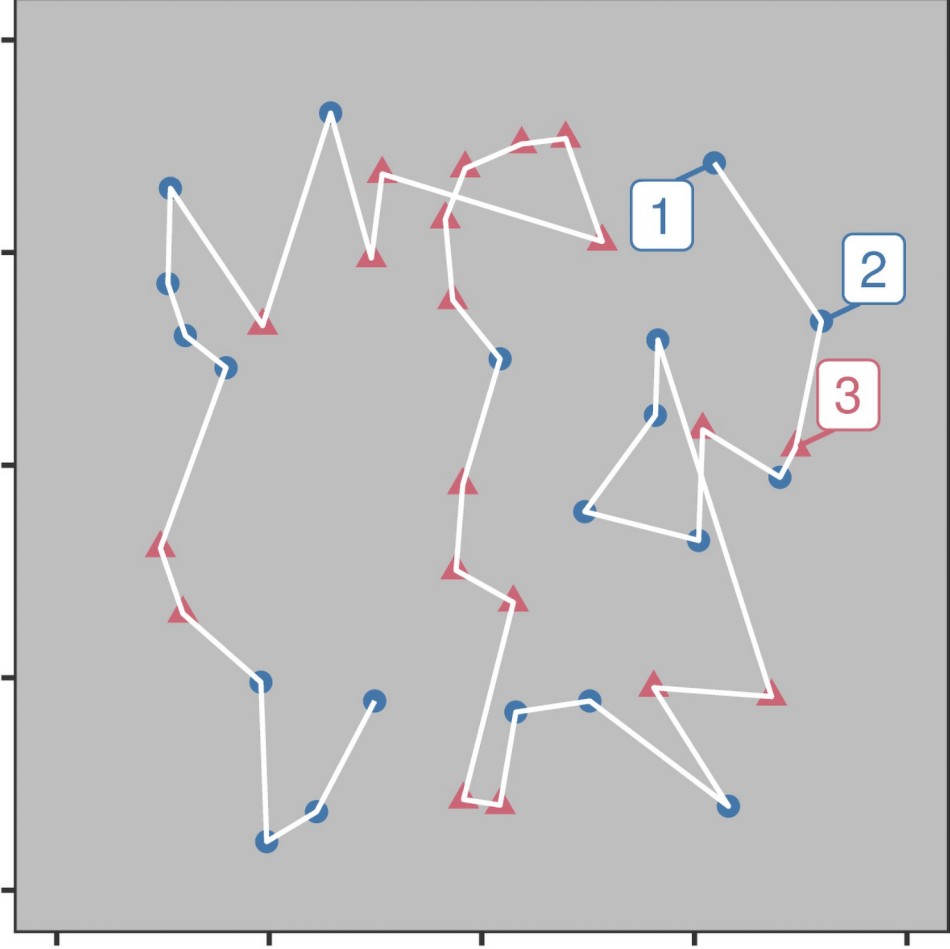

**Fig 2. An example of how foraging tasks can be thought of as spatial, using example data from Kristjánsson et al. (2014).** The 'path' taken by the forager is indicated by the white line, with the numbers indicating the order of targets taken (first three selections only).

and how humans and other species adjust their strategies in a way that is sensitive to these constraints is a question with broad implications for understanding the efficiency of behaviour more generally.

When focusing on target switching, behaviour similar to that seen in nonhuman animal studies has now been replicated multiple times in human adult participants [17–24]: in a task where participants need to collect as many targets as possible (while ignoring distractors), they tend to switch target type regularly when the task is relatively easy (e.g. when the targets can be distinguished from the distractors by a single feature dimension, such as colour, in an analogous manner to feature search in classical visual search paradigms). However, when participants are completing a more difficult search task (e.g. a conjunction search, where the target can only be distinguished from the distractor using two or more feature dimensions) they tend to search in 'runs', exhaustively searching for one target type before switching to another.

Human foraging studies have also highlighted that there can be surprising heterogeneity between participants in their search strategies. Some participants have been termed 'superforagers', showing no difference in foraging strategy for feature and conjunction type searches [17, 18, 25], potentially implying that they are able to hold multiple target templates simultaneously in working memory. However, recent work has also suggested that this group of people may be better described as employing a different strategy; working memory performance is not a good predictor of these individual differences [26], and in fact, it may not be a stable trait. Instead, people may be able to flexibly adjust their strategy in accordance with task demands [25, 27, 28]. It has even been suggested that 'superforagers' is a misnomer, as these participants may show more errors than 'normal' foragers, and instead these strategies may be sub-optimal responses to the task demands [29]. Interestingly, children under the age of 12 find conjunction foraging very difficult, indicating the important role of executive functions in this task, and highlighting that our understanding of the cognitive processes underlying run behaviour remains incomplete [24].

## Problems with run statistics

An episode of foraging entails a sequence of target-detection responses which have both a spatial and a target identity dimension. With each target selection response in the sequence, the distribution of remaining targets changes on both of these dimensions. In this respect, foraging data presents a unique analytical challenge relative to single-target search.

To study target switching, many previous studies have used a set of similar, highly correlated dependent measures: the target switch rate, the average number of runs per trial, or the average run length per trial. An issue with all of these measures is that the ground truth of the search area will influence target switching; the relative proximity and number of targets of the other category makes switches more likely. This is problematic for comparing run statistics across different conditions or studies that vary the distributions of target locations and categories. More fundamentally, run statistics do not adequately represent the underlying cognitive processes that might be determining foraging behaviour. Ideally, we would to model behaviour with respect to the relative contribution of parameters such as the tendency towards target constancy or switching target type, in conjunction with possible biases for one target type over another.

Run statistics also leave aside the intrinsic spatial aspects of foraging tasks. The effect of the distribution of foraging targets relative to the forager is central to understanding foraging efficiency: for example, patch leaving is affected by distance between patches [3], and heavy bees have a stronger proximity bias than lighter ones [30]. Despite this, relatively little work

has looked at target switching in direct relation to the effect of target proximity (although see [31] for one recent approach). This is especially important given that the spatial distribution of targets varies not only between conditions and studies, but also within a single episode of foraging, as each consecutive target is found and removed and the distribution becomes sparser.

## Our proposed approach

We propose an alternative strategy for analysing visual foraging data, which involves modelling the process as a generative sampling without replacement procedure, implemented in a Bayesian multilevel model. A strength of our approach is that it leads to measurements (model parameters) that are independent of the number of targets presented on screen and the number of targets that the participants are tasked with finding. In addition, we can estimate the latent biases that led to the run statistics.

We begin by demonstrating the modelling approach for a single trial of visual foraging data from [11], a classic study from the animal foraging literature. From this single trial, we can estimate a parameter that represents the probability that a forager will stick to the same target type or switch to a different type. We will then extend the model to multiple trials and conditions, adding in a bias for target type, and show that our model is able to detect small differences in these parameters that would not be picked up using the dependent variables of maximum run length or number of runs. We then extend to multi-level models to incorporate individual differences, and demonstrate the use of these models using previous data from [32].

Finally, we present a version of the model that incorporates a proximity bias, allowing us to estimate how likely participants are to select a target close to the previously selected target. We use this to demonstrate how the proximity bias affects the calculation of the other biases using example data from [21].

## Pulling balls out of a bag

We are modelling visual foraging as a process of weighted sampling without replacement, in what we call a 'bag foraging model'. Targets are selected one at a time and we define $t(i) = t_i$ as the $i$th target selected in a trial, $i = 1, 2, . . ..$ Our model is binary: targets can be one of two classes, $t_i \in \{a, b\}$, with $n_a$ and $n_b$ counting the number of each class of targets that have not yet been selected. $n_T = n_a + n_b$ is the total number of eligible targets remaining in trial. We assume that any biases in target selection strategy are constant within, and over, trials.

### Single trial

**Materials and methods.** We start with only one trial of data and assume targets of class $a$ and $b$ are equally likely be selected. We model the preference, $p_s \in [0, 1]$, for selecting a target of the same class as the previously selected target. Higher values of $p_s$ will lead to longer runs with the same class of target being selected repeatedly, while values of $p_s < 0.5$ indicate a preference for alternating between target classes.

For the first target selection, $t_1$, the probability of selecting a target of class $a$, is simply the proportion of $a$'s present in the stimulus:

$$Pr(t_1 = a) = \frac{n_a}{n_a + n_b} \tag{1}$$

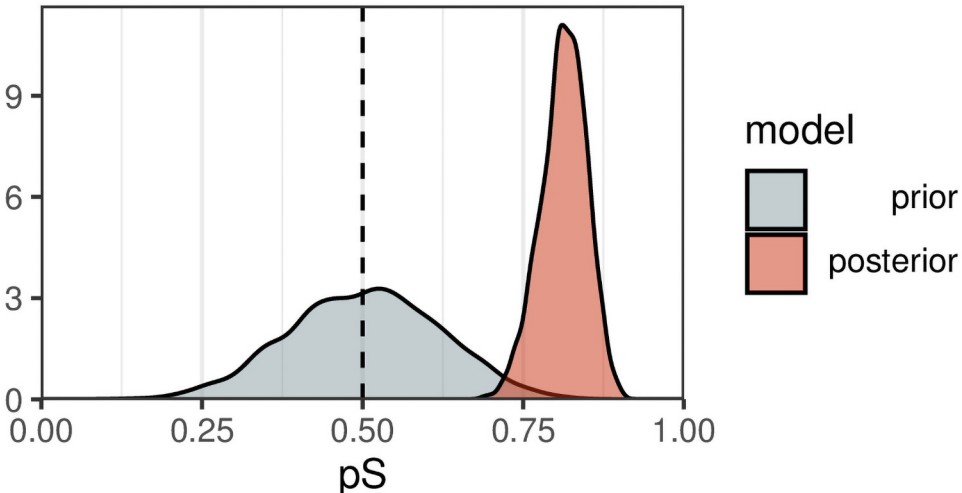

**Fig 3. Prior and posterior probability distributions for $p_s$ for the chicken foraging data from Dawkins (1971).**

For all subsequent target selections, $t_2$, $t_3$, ..., the probabilities $Pr(t_i = a)$ and $Pr(t_i = b) = 1 - Pr(t_i = a)$ depend on the the identity of the previously selected target, $t_{i-1}$:

$$Pr(t_i = a) = \begin{cases} \dfrac{p_s n_a}{p_s n_a + (1 - p_s)n_b} & \text{if } t_{i-1} = a \\[2ex] \dfrac{(1 - p_s)n_a}{(1 - p_s)n_a + p_s n_b} & \text{if } t_{i-1} = b \end{cases} \tag{2}$$

An easy way to think about $p_s$ is to consider the special case when there are equal numbers of both classes of targets left. In this case, $p_s$ can be interpreted as:

$$p_s = Pr(t_i = t_{i-1} | n_a = n_b) \tag{3}$$

The model was coded in Stan and we use logit link functions to ensure that our probabilities will lie between 0 and 1. I.e., we fit $b_s = \text{logit}(p_s)$. We define our prior as $b_s \sim N(0, 1)$, which translates to a prior prediction that $p_s$ will fall within 0.1 and 0.9 approximately 95% of the time. The code is presented in S1 Appendix and is available on the Open Science Framework (https://osf.io/7yuaz/). We ran the model using R (v4.0.3) and rStan, v.2.21.2, [33] with four chains and 1000 iterations. Further details of model checking procedures can be found in S1 File).

**Chicken foraging.** We imported chicken foraging data from [11], which is often referenced as one of the classic papers in the foraging literature. The data is from one chick, choosing between two types of food (grains of rice), and is shown in Fig 1.

After running the model on the data, we inspected traceplots and model summary statistics ($\hat{R}$, $n_{\text{eff}}$) to check that for successful convergence. These can be found in S1 File. The posterior is shown in Fig 3. By calculating the 53% and 97% HPDIs (highest posterior density intervals) around the preference to select the same target class, we obtain intervals of [0.80, 0.85] (more likely than not) and [0.73, 0.88] (very likely). We use these intervals over the more common, but equally arbitrary, 95% for the following reasons: i) 53% represents a little over half of the distributions while 97% represents most of it. ii) there are many cognitive biases associated with the interpretation of probabilities and it is possible that 50% intervals could be viewed as

being more likely to contain the "true" value than not. iii) 97% offers a more conservative choice than 95% while avoiding numerical difficulties in obtaining stable estimates out in the tails of the distribution.

**Stable estimates over trial length and number of targets.**   How much data is required to get a stable estimate of these biases? [11] only looked at the first 100 of 2000 targets found (5%). We ran a simulation where the chickens found greater proportions of the grains (10% or 100%) and found that our model does improve with more data, but only marginally: there is no great benefit to a 20-fold increase in grains in this simple simulation. We also ran a simulation where the chickens always found 100 grains, but the total number of grains varied, and again found this had only small effects on the estimates of $p_s$. Our method appears to offer robust model fits even with relatively small numbers of targets (see S1 File).

## Multiple trials & conditions

**Materials and methods.**   One of the issues with previous measures used in the foraging literature is that they are not able to address why run lengths may differ. That is, what are the underlying cognitive processes that drive a switch to a different target? Our method allows us to consider multiple biases: for example, in addition to a forager's tendency to stick with one target type or to switch ($p_s$) we can add in a second parameter, $p_a$, which reflects the bias towards one target type or another (e.g. if one target has a higher value). Increasing either or both parameters will lead to a smaller number of longer runs, but for different reasons.

We therefore expanded our model to work over multiple trials and conditions, and allowed participants to have a preference for one target class over the other:

$$p_a = Pr(t_i = a | n_a = n_b) \tag{4}$$

We estimate $p_s$ and $p_a$ independently for each condition, under the assumption that they are constant over trials from the same condition. Our model then becomes:

$$Pr(t_1 = a) = \frac{p_a n_a}{p_a n_a + (1 - p_a)n_b} \tag{5}$$

$$Pr(t_i = a) = \begin{cases} \dfrac{p_a p_s n_a}{p_a p_s n_a + (1 - p_a)(1 - p_s)n_b} & \text{if } t_{i-1} = a \\[3mm] \dfrac{p_a(1 - p_s)n_a}{p_a(1 - p_s)n_a + (1 - p_a)p_s n_b} & \text{if } t_{i-1} = b \end{cases} \tag{6}$$

See S2 Appendix for the Stan code; the model was run in a similar manner to the one trial model.

**Misattribution example: A $p_a$ bias with no $p_s$ bias.**   A strength of our model is it allows us to more accurately determine the biases that underlie foraging patterns. We can compare a condition with no biases ($p_a = 0.5$ and $p_s = 0.5$, a 'neutral' condition) with one where the observer prefers one target over the other ($p_a = 0.8$ and $p_s = 0.5$, which we call a 'target bias' condition, as in this case one target is preferred over the other e.g. because it has been given a higher value in the experiment, or because of its visual properties). An ANOVA (the standard statistical method used in foraging papers) can detect a statistically significant difference between these two conditions in terms of both the number of runs ($F = 91.1$, $p < 0.001$) and the maximum run length ($F = 99.1$, $p < 0.001$): see Fig 4A). However, it is not clear from this analysis what is driving these effects. The posterior probabilities from our model are presented in Fig 4C and clearly distinguish the underlying processes i.e. that there is no difference

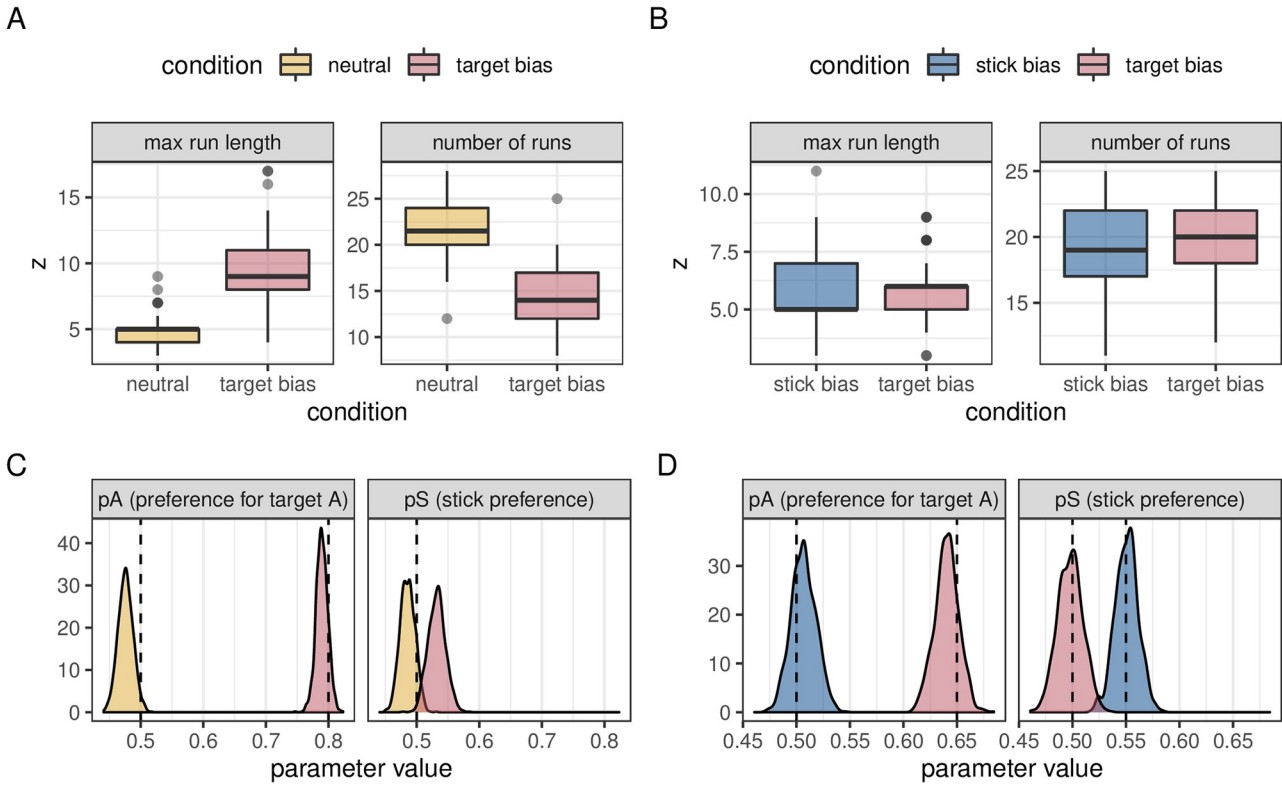

**Fig 4. Results for simulated biases in the bag model.** A: boxplots showing the maximum run length and number of runs in each of our two simulated misattribution conditions. B: boxplots showing the maximum run length and number of runs in each of our two simulated differentiation conditions. C: density plots showing the $p_a$ and $p_s$ values calculated by our model for each of the two simulated misattribution conditions. D: density plots showing the $p_a$ and $p_s$ values calculated by our model for each of the two simulated differentiation conditions.

between $p_s$ in the two conditions but a clear difference between $p_a$: if we repeat the procedure 100 times, we find a difference in $p_a$ 100% of the time, and a difference in $p_s$ only 2% of the time.

**Differentiation example: Two conditions which give the same run statistics.** To demonstrate the power of our approach, we can look at a simulation example, comparing a condition with a small stick bias ($p_a$ = 0.5, $p_s$ = 0.55) to a condition with a small target type preference ($p_a$ = 0.65, $p_s$ = 0.5). We simulated 50 trials from an experiment with these two conditions, with each trial consisting of 40 targets. As can be seen in Fig 4B, these two conditions give rise to very similar distributions of maximum run lengths and number of runs. Indeed, when applying the standard ANOVA to these data, we fail to detect a statistically significant difference: for example, in one simulated dataset, $F$ = 0.02, $p$ = 0.9 for the maximum run lengths and $F$ = 0.41, $p$ = 0.52 for the number of runs.

In contrast, running our model on these data results in the posterior probability distribution seen in Fig 4D. From these, we can calculate the difference between conditions, which can be summarised with 97% HPDIs for $p_s$, [−0.09, −0.02], and $p_a$, [0.10, 0.17]. These intervals do not cross 0, consistent with a difference between conditions for each measure separately. An alternative way to summarise these posterior distributions is to calculate $Pr(\text{difference} > 0|$ data). We can then carry out a power analysis by repeating this procedure 100 times. When we do this, we detect a significant difference in $p_a$ and $p_s$ in 100% and 83% of simulations,

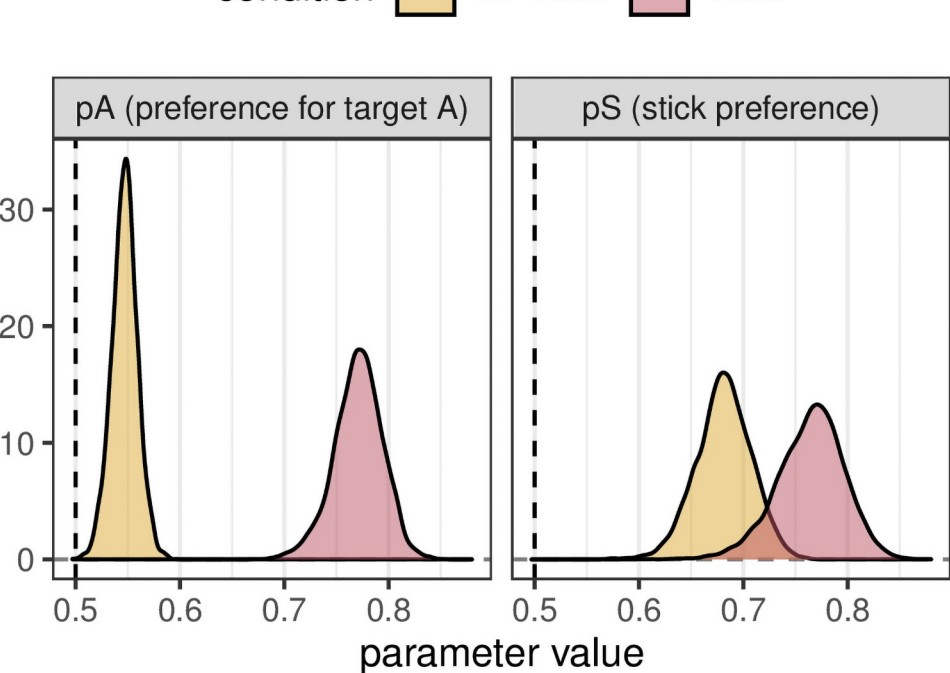

**Fig 5. $p_a$ and $p_s$ biases across participants in multi-level re-analysis of Tagu and Kristjánsson (2021), showing both the value and no-value conditions.**

respectively. In contrast, applying the same power analysis using the standard run statistics, we find a statistically significant difference in the number of runs in just 8% of the simulations. Similarly, we find a difference 21% of the time for the maximum run length statistic. We detect a difference in either statistic, and hence detect a difference between conditions, only 29% of the time. Our approach is therefore not only more sensitive than run statistics to the difference between conditions, but is able to correctly attribute the difference to a combination of a bias to switch and a preference for one target over the other.

**Multi-level: Tagu and Kristjánsson (2021).** The final version of the bag foraging model allows multiple participants in a multi-level framework (see S3 Appendix). Both $p_s$ and $p_s$ are allowed to vary from one participant to the next.

Using simulated data, we can show that our model can recover both overall $p_s$ and $p_a$ biases, as well as the parameters from individual participants (see S1 File for details).

Our model also works well with more complex examples. In a recent study, [32] included a target value manipulation, where participants searched for three target types (among three different types of distractors). Participants had to earn a certain number of points in order to complete the trial, and in the 'value' condition, one of the target types was deemed high value, whereas the other two were low value. We would therefore expect this condition to have a higher $p_a$ bias than the 'no value' control, where all targets had the same value. This is indeed what we see in Fig 5. This example also shows the flexibility of the model; in this experiment, the proportion of the different target types (high and low value) was not equal to 50:50, and trials were not necessarily equal lengths. Note, for simplicity, we considered only the blocks where participants used a computer mouse in our analysis, and only the trials where the targets had a random distribution (see S1 File).

## Discussion

The bag foraging model we have presented above can account for run behaviour in foraging tasks. Alongside data from [11] and [32], we have also fit the multi-level model to data from [17] (presented in S1 File) and can replicate their key findings. For example, on average, participants have a lower $p_s$ in the feature condition than in the conjunction condition, demonstrating that they are more likely to have longer 'runs' of one target type in the conjunction condition. We can also clearly identify the so-called 'superforager' participants, who show a much smaller gap between conditions. However, our method also allows us to see new patterns in the data. For example, the $p_a$ values in the conjunction condition were often below 0.5. This indicates that participants had a preference for the targets labelled 0, suggesting that by chance, in this dataset, these targets were more salient: S1 File shows how the individual $p_A z$ correlated well with the participants' tendency to select a '0' for the first target on each trial. Our method therefore allows deeper analysis of the underlying biases than has been possible with the metrics used in previous studies, while also correlating well with those measures.

One limitation of our model is that it can give less accurate parameter estimates if there are very extreme biases e.g. if the two target types are found almost exclusively in disjoint, non-overlapping runs. If $p_s = 1$ (and $p_a = 0.5$), our simulated results suggest that the recovery of the $p_s$ parameter will be accurate, but the posterior for $p_a$ is shifted away from 0.5. If we take the converse example where $p_a = 1$ (and $p_s = 0.5$), we obtain a model fit that generates posterior predictions that are identical to the types of runs that would be generated by these parameters i.e. the model correctly predicts that all the targets of one type will be selected first. However, we see that both $p_a$ and $p_s$ estimates are actually close to one (see S1 File). This is because an extremely strong target preference means that the participant will find all these preferred targets before the non-preferred, generating a trial where they appear never to switch. Thus, the estimated parameters are entirely consistent with the observed data. However, assuming a slightly less extreme value of $p_a = 0.97$, or adjusting the priors, allows better recovery of the correct parameters (see S1 File). In the case of the [17] data, the $p_a$ bias appears to be real, as manual inspection of the first five target selections per trial in the conjunction condition shows a similar bias towards the target labelled '0'. Thus, some caution should therefore be exercised in interpreting parameters where there are strong biases, but it is simple to check if there are model fitting problems.

## Spatial biases

### Materials and methods

Our initial model incorporates several biases which appear to help explain the findings in previously published research [32]. However, this model has no representation of the spatial location of targets. It seems plausible that people are more likely to select targets nearby to the one they have just selected. Proximity could be particularly important in feature search, where the targets are relatively easily distinguished so there are fewer additional constraints on selection; if so, perhaps proximity is relatively less important for conjunction search. We can also investigate the effect of direction/momentum: do observers incorporate some form of momentum to their target selections, or do they prefer to double back on themselves? Extending our bag foraging model to incoporate spatial biases allows us to see to what extent these hypotheses are true, and how the inclusion of a spatial bias will affect our estimation of the other biases.

For our second model, we take our stimuli to be a collection of targets, $t(i) = (t_i, x_i, y_i)$, where $(x_i, y_i)$ gives the target's location and $t_i \in \{a, b\}$ gives the target's class. For mathematical convenience, we will take $a$ and $b$ to factor levels with $a = 1$ and $b = -1$. If we take $d(i, j)$ to be the Euclidean distance between targets $(x_i, y_i)$ and $(x_j, y_j)$, we can define a proximity measure

between the two items as:

$$\rho_d(i,j) = e^{-\sigma_d d(i,j)}$$

where $\sigma_d$ is a proximity tuning parameter that controls how quickly $\rho$ decreases from 1 to 0.

Similarly, we can define the angular difference $\theta(i,j)$ to be:

$$\theta(i,j) = \frac{f(\text{atan2}(i,j) - \text{atan2}(i-1,i))}{\pi}$$

where atan2$(j, i)$ is the direction of travel from $i$ to $j$ and $f(\phi_1, \phi_2)$ calculates the angular difference:

$$f(\phi_1, \phi_2) = \min((\phi_1 - \phi_2)\%2\pi, (\phi_2 - \phi_1)\%2\pi)$$

This should give a value between 0 and 1, with 0 indicating a target that "in front" of the current target, 0.5 represents a change of direction of 90˚ and 1 represented doubling back in the direction we just came from. Having calculated $\theta$, we can now define:

$$\rho_\theta(i,j) = e^{-\sigma_\theta \theta(i,j)}$$

In the bag foraging model, all targets of the same class were interchangeable and so a simple Bernoulli process could be used to model each target selection. However, this no longer holds, as each target has a unique spatial location that needs to be taken into account. Therefore we will use a categorical model with the weights for each remaining target being defined as:

$$w(i) = g(b_a t_i + b_s m(t_i, t_{i-1})) \times \rho_d(i, i-1)\rho_\theta(i, i-1) \tag{7}$$

where $g$ is the inverse logit function and $m$ is an indicator function that equals 1 if $t_i = t_{i,1}$ and -1 otherwise. Targets that have already been selected earlier in a trial have their weights set to 0. S4 and S5 Appendices show the code for the basic and multi-level spatial models, respectively.

## Results

**Simulation examples.** We present a simple example using simulated data in S2 File, and demonstrate that we can recover the parameters used to generate the data. A more interesting example can be seen in the case where the targets have spatial structure in the environment, being found in patches (See S2 File for example stimulus). In our simulation, we set $b_a$ and $b_s$ to be zero, and used a proximity tuning value of 15. If analysed without including a proximity bias parameter, the model mistakenly generates a $b_s$ bias: the clumping of targets does indeed lead to them being collected in runs, simply because our simulated participant has a preference for nearby targets (see Fig 6). However, the model including the proximity bias parameter is able to recover the correct parameters. Our model is therefore able to distinguish between a preference for a *similar* target and a preference for a *nearby* target, a distinction that is likely to be particularly important in more naturalistic set ups where target clumping might be expected to be prevalent.

**Multi-level: Clarke et al. (2018).** The data from [21] closely replicates [17], yet has around four times as many participants, giving us greater power to investigate individual differences in feature and conjunction foraging. The fixed-effects of the model are shown in Fig 7. There is a small bias towards repeating target selections during feature search, and this increases to approach ceiling in the conjunction condition. There is also a very strong proximity bias in both conditions, with very little weight given to targets more than a quarter stimulus width from the currently selected target. The parameter for the direction bias comes out as negative, leading the model to favour doubling back on itself to select targets that may have been skipped over (see S2 File for simulated examples of how different direction biases affect foraging behaviour).

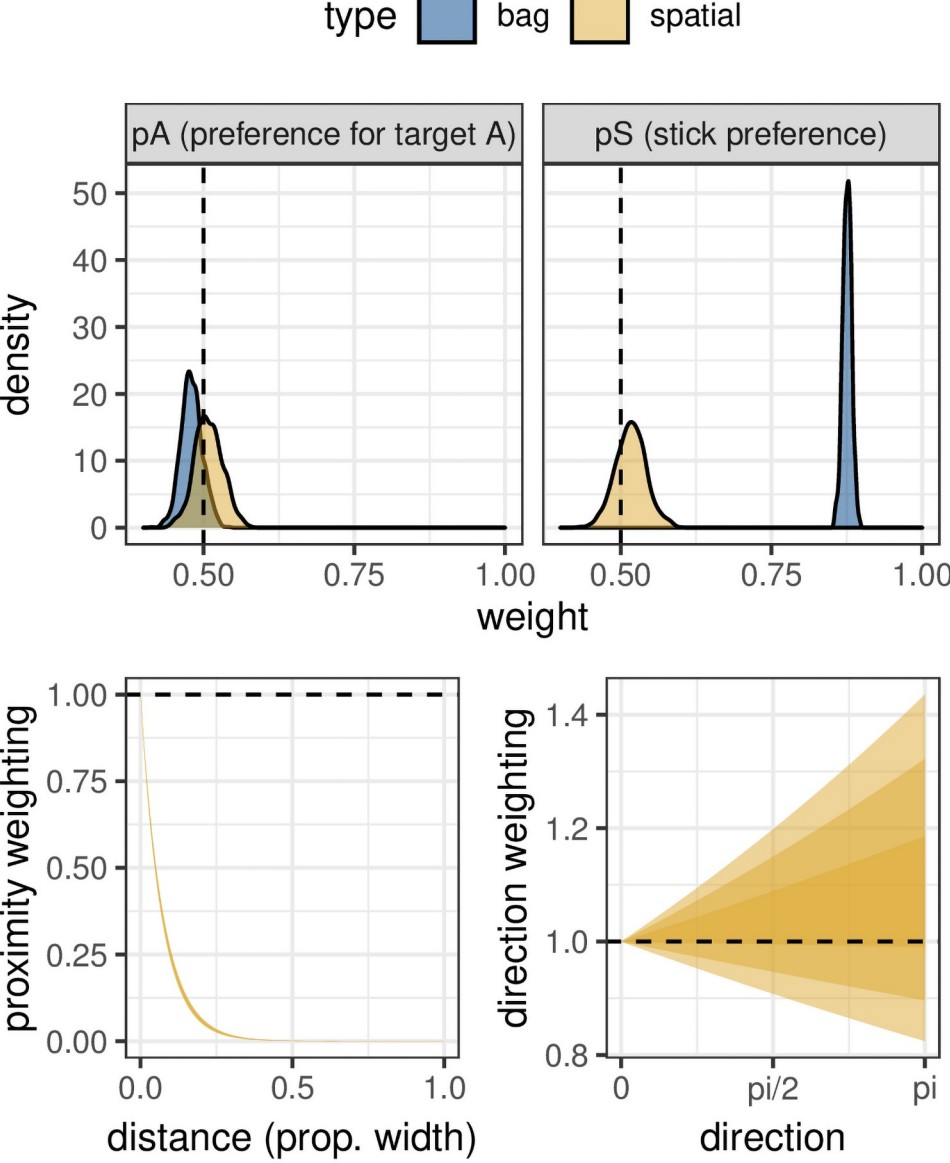

**Fig 6. Posterior distributions for both a bag foraging model and a spatial model trained on patchy stimuli, where target types are clumped.** The shaded ribbons for the proximity and direction weighting indicate 53%, 89% and 97% HPDIs. The spatial model gives more weight to nearby targets (i.e. where the distance is close to zero), and has no particular preference for any direction, as indicated by the fact that a value of 1 falls within the HPDI for all directions.

Interestingly, both of these biases are slightly weaker for conjunction foraging, suggesting participants are relying less on spatial attention and emphasising feature-based attention.

As this model was fit using a multi-level framework, we can also look differences between participants. Fig 8 shows how the parameters vary over participants and conditions. We can see that with the exception of parameters linked to the spatial biases, there are little-to-no correlations between different parameters across conditions, or within parameters across conditions. However, the two spatial parameters appear to be correlated: participants with a stronger preference to select nearby targets also have a more negative direction bias. Participants with a weaker proximity bias are more likely to have a positive directional bias. While there is no clear partition of participants into subgroups, the two opposing ends of this range

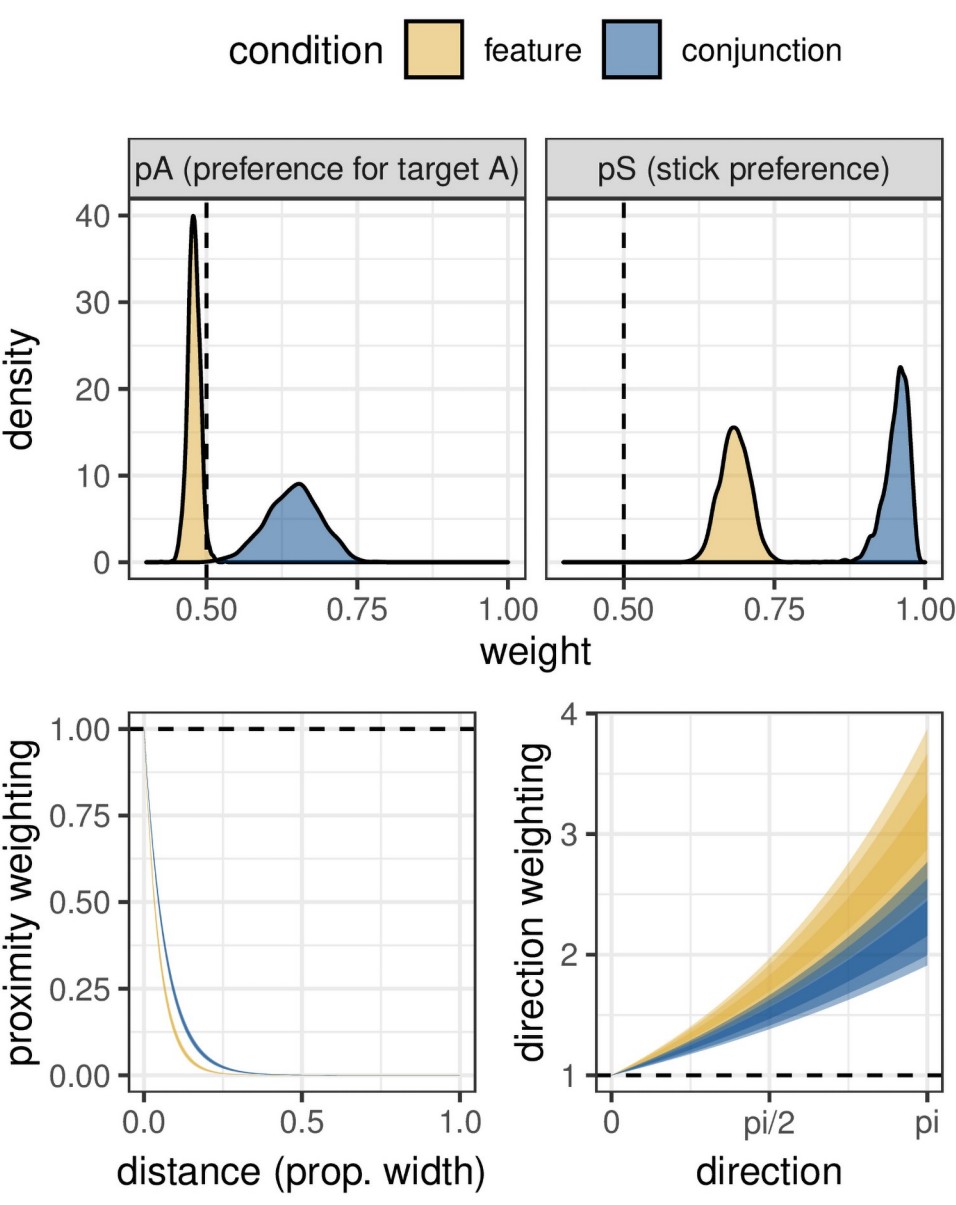

**Fig 7. Posterior distributions for our model when trained on data from Clarke et al. (2018).** The shaded ribbons for the proximity and direction weighting indicate 53%, 89% and 97% HPDIs. Note that there is a higher weighting for directions closer to pi i.e. completely reversing the direction, indicating negative momentum.

could be interpreted as a strategy to search within local patches compared to a more global scanning strategy.

Finally, we can use the fitted model to simulate new data and compare run statistics and inter-target distances with the original empirical data. We do this for $n = 100$ samples from the fitted posterior probability distributions so that we can quantify the uncertainty in the model's predictions. The results (Fig 9) show a good correspondence between the posterior predictions and empirical data. Both the summary run statistics (the two statistics most commonly analysed in the visual foraging literature) are a close-to-perfect match between the model and the empirical data. The inter-target distances are more interesting. The model offers a reasonably good fit, especially in terms of the mean distances and in the later half of a trial. However, we

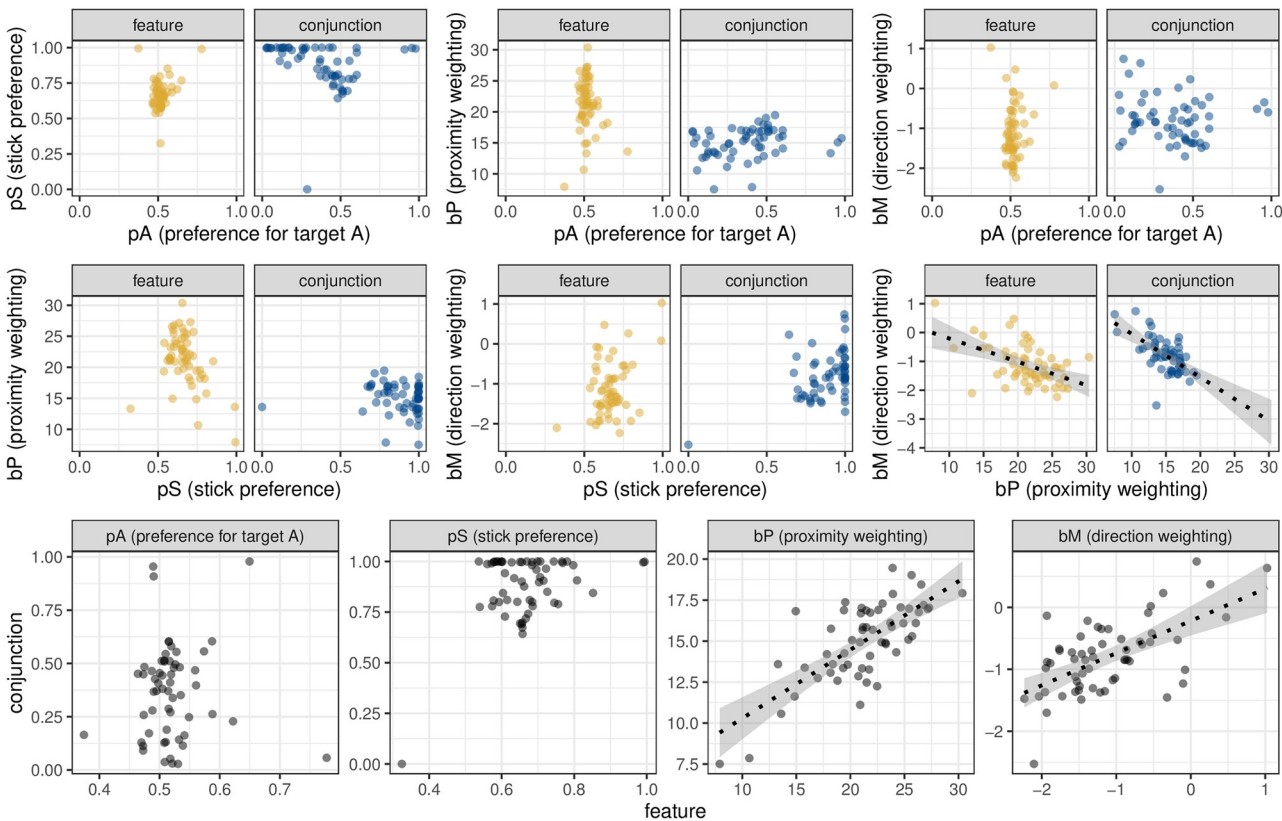

**Fig 8. Individual differences in Clarke et al. (2018).** The first two rows show the correlations in the random effect structure between different parameters. We can see that $p_s$ and $p_s$ appear to be independent from one another, and independent from the spatial biases $b_p$ and $p_m$. However, $b_p$ and $p_m$ are correlated with one another in both the feature and conjunction conditions. The bottom row shows the correlation between conditions for each parameter in our model.

can see two main differences: i) the variance in the human data is larger, suggesting that perhaps the negative exponential proximity function down-weights more distant targets too much; and ii) there is an interesting dynamic at the start of the trial in which the first few targets selected tend to have shorter inter-target distances than the model would expect. This raises the possibility that participants either chose an initial target that is in a relatively dense patch of potential targets, or that they are able to accurately plan out an efficient order for the first five or so target selections that minimises the distance that they would be required to travel. This observation highlights another use of the model, which is to detect deviations in human behaviour from the model's predictions. These deviations can guide the direction of further investigation.

## Discussion

The spatial model presented extends the simple bag foraging model to incorporate new parameters that further explain participant behaviour on foraging tasks by accounting for spatial attention. This section demonstrates the flexibility and modularity of the bag foraging model, which can be used as a starting point for exploring many questions about foraging behaviour. As well as presenting the model fit for the [21] dataset above, S2 File presents fits for a number of other datasets, including [17], [32] and [28]. Another key strength of our model demonstrated in S2 File is that it is able to generate good parameter estimates with relatively little data (e.g. one trial per participant per condition), in stark contrast to using traditional run statistics,

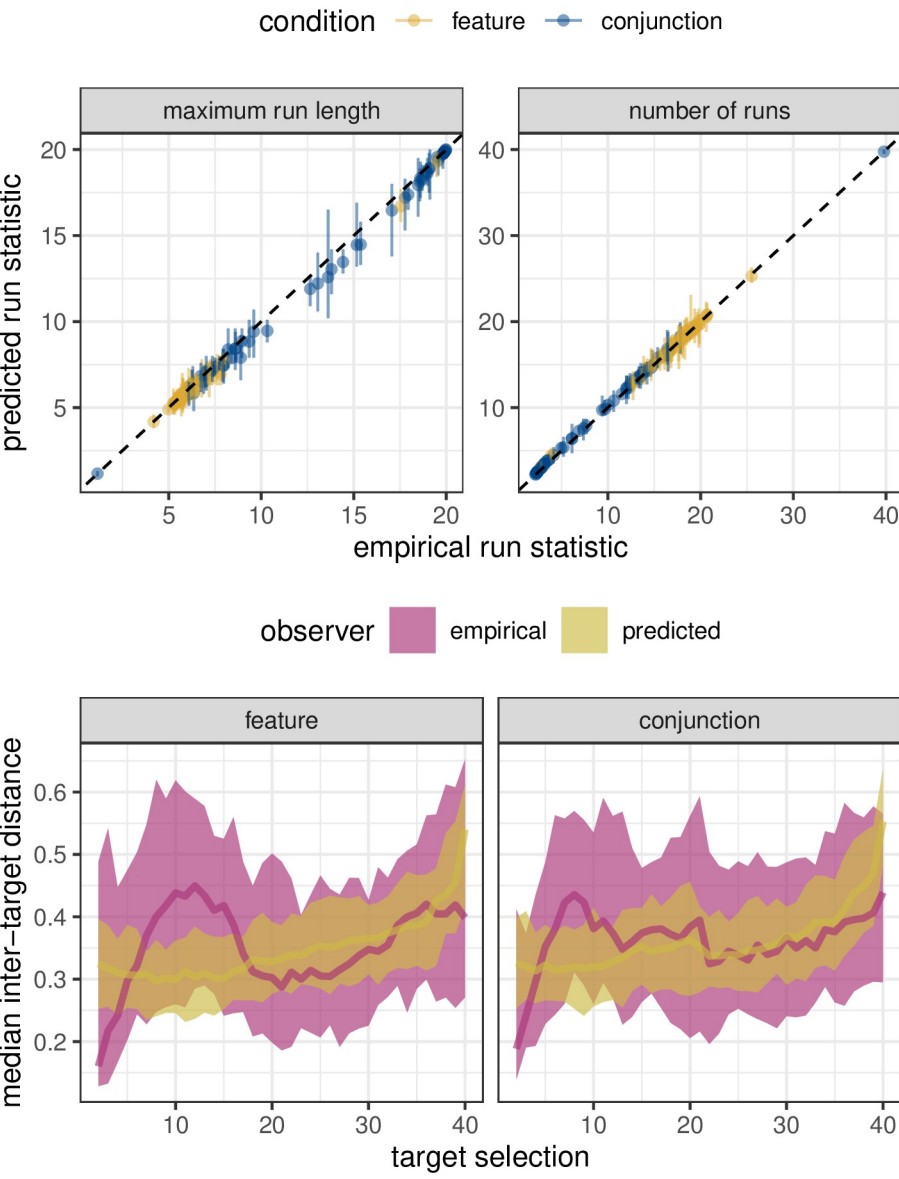

**Fig 9. Posterior predictions for Clarke et al. (2018).** (*top*:) Scatter plots between empirical and predicted summary run statistics. Error bars indicated 89% HPDI for the posterior predictions. (*bottom*:) Comparison of inter-target distances and how they vary over time. Shaded region gives 89% intervals.

which reduces each trial to one summary number. This means that our model could allow analysis of foraging data from populations where it is difficult to carry out large numbers of trials, such as from clinical populations or when studying animal behaviour.

## General discussion

One of the biggest strengths of our approach is its relative simplicity. By taking the task structure of sampling without replacement into account, we need only four parameters to characterise the relatively complex sequences of runs and inter-target distances produced during foraging tasks. While approximating these parameters with summary statistics (e.g. the mean of the run lengths) in many cases will lead to broadly similar conclusions, we have

demonstrated that in some instances, run statistics give misleading results that our approach avoids. In addition, summary statistics require much more data per participant and condition to be able to make good estimates. By treating each selection instance as a separate observation, our approach reaches stable estimates of biases far more efficiently.

Our model is also extremely flexible, and could be adapted in many ways. The underlying assumptions of the model are all tuneable and testable: for example, it would be straightforward to compare and contrast different proximity functions, as our choice of using an isotropic negative exponential is relatively arbitrary, and other functions may offer improved performance for predicting behaviour. The momentum function also assumes a monotonic trend from going forwards in a straight line to going backwards, but again, there are other possibilities that could easily be tested. For example, it could also be explored whether functions that allow for a forward-backward preference over perpendicular movement are better. Interesting further questions could explore momentum and proximity biases across different modalities. Most of the data included here was based on finger or mouse-click foraging, but larger movements (such as moving around a large area) or smaller ones (such as eye movements) are likely to be characterised by different proximity and momentum functions.

More broadly, the model is extremely straightforward to extend. As it stands, our model works well with different types of foraging tasks, as we have demonstrated using both exhaustive and non-exhaustive foraging example datasets. However, while outside the scope of the current paper, it would be easy to also implement stopping rules for models of non-exhaustive foraging, and it would be possible to make quantitative predictions about distributions by assuming different stopping rules (for example, by assuming that after each target selection, there is some given probability to terminate the trial). This would allow modelling of patch-leaving rules, assuming that one trial acts as one patch. Within a trial, it would also be possible to model patches made up of multiple "clumped" targets on a single large display, by modifying the spatial bias component to include a categorical factor to indicate the different clumps of targets (that are therefore assumed to be perceptually 'further away' from each other). The $p_a$ bias could also be broken down into sub-biases based on how preference changes based on stimulus properties such as colour, luminance and value. The model could also be extended to explore how biases change temporally (e.g. over multiple trials), or to include new biases, such as a bias for the initial target selection. Finally, it would also be possible to extend beyond two target categories: for example, for three categories, instead of fitting one value for $Pr(t_i = a)$ and and then letting $Pr(t_i = b) = 1 - Pr(t_i = a)$, we would need to add an extra parameter, which could be parameterised $Pr(t_i = c) = 1 - Pr(t_i = a) - Pr(t_i = b)$.

In other aspects of foraging, such as the marginal value theorem for predicting when foragers should leave a patch [9], formal mathematical modelling has been used to effectively represent the underlying structure of foraging behaviour. Our work provides a similar framework for target selection during foraging, extending our theoretical models of foraging behaviour and giving a way for researchers to begin to understand the cognitive processes that underlie performance in these tasks. We envision future extensions bringing these and other models together into a complete understanding of all aspects of foraging. While a full account of human foraging is of course likely to be extremely complex, involving many factors [10], we are optimistic that our approach will provide a framework to begin this challenging task. By making our code freely available, we hope to enable other researchers to update the model as they require, perhaps to take into account other cognitive factors that they think may be important in foraging tasks, or to enable them to use the model to study their own variants of the task.

There are a number of potentially important factors that we do not consider in the current model. For example, it does not currently allow for modelling inter-target selection times, which are thought to be important in determining foraging behaviour [28]. Similarly, it does

not explicitly model the effect of background clutter or distractors, although implicit effects (such as decreasing the salience of one of the target classes) will be captured. As most foraging studies to date have not focused on distractor effects, information on the distribution of distractors is not routinely included in freely-available datasets, making it more difficult to model this aspect of the task. However, if such data were available, the model could be extended to treat distractors in the same way as targets, but with a sampling-with-replacement and inhibition of return process for when a distractor is selected.

## Conclusion

Our Bayesian statistical model of visual foraging provides a simple and flexible way to estimate cognitively meaningful parameters underlying foraging tasks. We demonstrate that the model is able to reproduce the patterns found in existing data, and also provides new avenues for research questions, such as the role of salience in predicting target preferences and how proximity and direction biases may represent stable individual differences across different task variants. By considering the task structure in developing our model, we can go beyond simple linear models to generate more powerful predictions and deeper insights into foraging behaviour.

## Supporting information

**S1 File. Supplementary materials part 1.** Supplementary materials for the bag foraging model.
(PDF)

**S2 File. Supplementary materials part 2.** Supplementary materials for the spatial foraging model.
(PDF)

**S1 Appendix. Bag foraging model—One trial.** The Stan code for the basic version of our sampling-without-replacement model for visual foraging (may be opened in R or in a text editor).
(STAN)

**S2 Appendix. Bag foraging model—Multiple trials and conditions.** The Stan code for the version of our sampling-without-replacement model for visual foraging, incorporating multiple trials and conditions (may be opened in R or in a text editor).
(STAN)

**S3 Appendix. Bag foraging model—Multi level.** The Stan code for the version of our sampling-without-replacement model for visual foraging, incorporating a multi-level random effects structure (may be opened in R or in a text editor).
(STAN)

**S4 Appendix. Spatial foraging model.** The Stan code for the version of our sampling-without-replacement model for visual foraging, incorporating spatial biases (may be opened in R or in a text editor).
(STAN)

**S5 Appendix. Spatial foraging model—Multi level.** The Stan code for the version of our sampling-without-replacement model for visual foraging, incorporating spatial biases and a multi-level random effects structure (may be opened in R or in a text editor).
(STAN)

## Acknowledgments

The authors would like to thank all researchers who share their data on OSF.

## Author Contributions

**Conceptualization:** Alasdair D. F. Clarke, Amelia R. Hunt, Anna E. Hughes.

**Data curation:** Anna E. Hughes.

**Formal analysis:** Alasdair D. F. Clarke.

**Funding acquisition:** Alasdair D. F. Clarke, Amelia R. Hunt.

**Methodology:** Alasdair D. F. Clarke.

**Resources:** Alasdair D. F. Clarke.

**Software:** Alasdair D. F. Clarke.

**Visualization:** Alasdair D. F. Clarke, Anna E. Hughes.

**Writing – original draft:** Alasdair D. F. Clarke, Amelia R. Hunt, Anna E. Hughes.

**Writing – review & editing:** Alasdair D. F. Clarke, Amelia R. Hunt, Anna E. Hughes.

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
