## [Decision Letter · Decision Letter 0]

6 Dec 2021

Dear Dr Hughes,

Thank you very much for submitting your manuscript "Building Bayesian Cognitive Models of Visual Foraging" for consideration at PLOS Computational Biology. As with all papers reviewed by the journal, your manuscript was reviewed by members of the editorial board and by several independent reviewers. The reviewers appreciated the attention to an important topic. Based on the reviews, we are likely to accept this manuscript for publication, providing that you modify the manuscript according to the review recommendations.

As you will see from the comments below the reviewers were all very positive and appreciative of the effort put into thie work. We expect this to be a minor revision.

Sincerely,

Ulrik R. Beierholm

Associate Editor

PLOS Computational Biology

Natalia Komarova

Deputy Editor

PLOS Computational Biology

[LINK]

Reviewer's Responses to Questions

**Comments to the Authors:**

Reviewer #1: The paper Building Bayesian Cognitive Models of Visual Foraging by Clarke et al. introduces a new approach to modeling data from visual foraging tasks. Importantly, the approach can disentangle different influences on target selections and predict e.g. run-behavior in foraging for different conditions. The model is a much-needed tool for a field that lacks adequate methods to deal with the rich data produced by foraging tasks.

The paper is well structured, the rationale is clear, and the model and its applications are presented in a comprehensible manner. I only have a few minor points and suggestions, which I list below (not necessarily in order of importance).

* p2: "... below the average rate of return across all patches". - I suggest adding something to highlight that the avg. ROR across all patches also includes travel time during which no elements can be collected. Sometimes people not familiar with MVT are confused about this ...

* p2: The idea of a "search image" is used on p2. Perhaps after this passage, you could mention the related concepts of attentional templates concerning visual search in humans. The concept of templates appears later in the current manuscript but was nowhere introduced ... this might be a possibility.

* On p2 you suggest that foragers can keep "multiple target templates active in working memory". On p3 you write " potentially implying that they are able to hold multiple target templates simultaneously in working memory", as a special case for super-foragers, implying that normally this might not be possible (somewhat in conflict with what was said on p2). I think the topic is still debated. The point here could probably be made without committing to either view.

~ p4: It does not become clear that (and why) the model focuses on a binary case with two target types. Also, in the discussion, where limitations are discussed, this point is not taken up either.

* On several occasions, you point out that existing methods might not adequately model the underlying cognitive processes, implying that your approach does. At one location, it is also explicit: "A strength of our model is it allows us to more accurately determine the cognitive processes underlying foraging patterns." While I completely agree that your model is a significant step forward compared to earlier methods and goes a long way in disentangling different biases, I would say that it still does not "determine the cognitive processes underlying foraging patterns." How do these processes lead to a particular switching frequency? Why is there a preference for certain features? ....

* p8: Suddenly, the term "bag model" appears. It also occurs as "bag foraging model", sometimes set in italics. Perhaps this term should be introduced earlier.

* p5: Figure 3 refers to "salience" (and it comes up later again). The main text does not introduce the idea that salience could drive a target tepy preference. The real-data example uses the value as the reason for preferences ...

* p5: Figure 2: "the chicken data" sounds a bit odd. Perhaps add a reference?

* p7: I find the term "conflated power example" unclear.

* p9: "identical to the training data": I think the ML framing (which only appears here) could be confusing for some readers

* p11: "leading the model to favor doubling back on itself to select targets.". I have only a vague idea of what this means ... perhaps a figure could be helpful here.

* p11: what is an "eligible target". Aren't all targets eligible?

* p15 (an also the title): "Bayesian cognitive model" - I know this term is used for different things (and e.g. in the Lee and Wagenmakers book in a similar context as here). Personally, I try to avoid it because people easily confuse it with Bayesian models of cognition, i.e. the idea that cognitive processes follow Bayesian principles. If I see it correctly, your study does not suggest this (of course, it might also not exclude it) but simply uses a Bayesian statistical framework.

* The paper contains many typos. Here are some I spotted:

- p1, 1st para: missing closing parenthesis

- Figure 1: Caption says "top" and "bottom," but panels are on the left and right.

- Figure 1: I think "first three trials" should be "first three selections"

- p5: HDPIs  HPDIs, also the acronym is never introduced; in Figure 5 you use "HDI".

- p5: Footnote: consider removing point iv). People not very familiar with Bayesian stats might think this might really be of relevance

- p6: "(5%" - closing parenthesis missing

- p6: "our model__s__ then becomes"

- p7: "seen in 3" ... "Figure" is missing

- "super-foragers" vs. "superforagers"

- P10: "Figure Figure 5"

- p11: "to appear to be"

Reviewer #2: I was impressed with this manuscript. It is written very clearly and efficiently and presents to me what seems like a very useful approach to understanding foraging. While I do think that accounts based on run behavior can still definitely aid our understanding of foraging in particular and visual attention and attentional selection more generally, the current approach has considerable merits, which the authors make a very clear and concise case for.

I therefore do not have many comments, and my comments therefore can be taken as food for thought.

Firstly, I would recommend not talking about "confounds" with regard to the literature that has focused mostly on selection runs. Yes, it is true that run length, run number, switch rates are interdependent and correlated, but I do not see the problem that the word confound implies. As I see this, the current authors sampling approach can well hold it’s own without trying to imply that run based approaches need to be avoided as language such as confounds certainly implies. The uncertainty with regard to cognitive mechanisms behind run behavior is something that the authors could focus more in in my opinion – for example the finding that children under 12 tend not to switch, may point towards mechanisms.

When the authors talk about how optimal foraging theory has sometimes run into problems, I thought they should mention a recent finding by Kristjánsson, Björnsson & Kristjánsson (https://doi.org/10.3758/s13414-019-01941-y ) where the authors claim that human foraging in paradigms in many ways identical as those under investigation in the manuscript here deviated from the predictions of optimal foraging conceptions and that participants kept on foraging within the same patch much longer than expected. The authors concluded that human foraging is probably influenced by too many factors to be captured with a relatively simple mathematical model. This actually speaks to what my main point about this paper is in that there are so many factors at play that determine foraging performance that capturing them all within a single model is probably premature, if not impossible.

With regard to this issue of „patch leaving“, I encoourage the authors to provide a little more detail and be a bit more explicit in the discussion about how this could be implemented within their model. This is only briefly metnioned in the current discussion.

I went through the mathematical modelling and it seemed to be very solid.

I do share the authors enthusiasm with regard to cluttered scenes. It would for example be highly interesting to vary noise levels – and in fact many means are possible it seems.

Overall, I commend the authors on what is in my opinion a very fine contribution to the literature.

I was impressed with this manuscript. It is written very clearly and efficiently and presents to me what seems like a very useful approach to understanding foraging. While I do think that accounts based on run behavior can still definitely aid our understanding of foraging in particular and visual attention and attentional selection more generally, the current approach has considerable merits, which the authors make a very clear and concise case for.

I therefore do not have many comments, and my comments therefore can be taken as food for thought.

Firstly, I would recommend not talking about "confounds" with regard to the literature that has focused mostly on selection runs. Yes, it is true that run length, run number, switch rates are interdependent and correlated, but I do not see the problem that the word confound implies. As I see this, the current authors sampling approach can well hold it’s own without trying to imply that run based approaches need to be avoided as language such as confounds certainly implies. The uncertainty with regard to cognitive mechanisms behind run behavior is something that the authors could focus more in in my opinion – for example the finding that children under 12 tend not to switch, may point towards mechanisms.

When the authors talk about how optimal foraging theory has sometimes run into problems, I thought they should mention a recent finding by Kristjánsson, Björnsson & Kristjánsson (https://doi.org/10.3758/s13414-019-01941-y ) where the authors claim that human foraging in paradigms in many ways identical as those under investigation in the manuscript here deviated from the predictions of optimal foraging conceptions and that participants kept on foraging within the same patch much longer than expected. The authors concluded that human foraging is probably influenced by too many factors to be captured with a relatively simple mathematical model. This actually speaks to what my main point about this paper is in that there are so many factors at play that determine foraging performance that capturing them all within a single model is probably premature, if not impossible.

With regard to this issue of „patch leaving“, I encoourage the authors to provide a little more detail and be a bit more explicit in the discussion about how this could be implemented within their model. This is only briefly metnioned in the current discussion.

I went through the mathematical modelling and it seemed to be very solid.

I do share the authors enthusiasm with regard to cluttered scenes. It would for example be highly interesting to vary noise levels – and in fact many means are possible it seems.

Overall, I commend the authors on what is in my opinion a very fine contribution to the literature.

Reviewer #4: This paper describes two versions of a model of visual foraging aiming at providing more details about foraging behaviour than the run statistics often used in the literature. Notably, run statistics can inform about the tendency to stick with one target type or to switch, while the “pulling balls out of a bag” model offers the possibility to assess pA (the preference for one target type or another) on the top of pS (the tendency to stick with one target type or to switch). The second version of the model allows assessment of spatial biases with two additional parameters for proximity and direction.

I enjoyed reading this paper. It is nicely written, and the paper is well motivated and of high interest in the field of visual foraging. The maths is explained in a very pedagogical way so that it is also understandable by non-experts. Moreover, I wish to thank the authors for providing useful, detailed and well-constructed supplementary materials on the top of the main article.

I only have a few comments below that I hope are useful.

Jerome Tagu

1. I find the four parameters described in the paper useful for assessing foraging behaviour, but I feel they are presented independently, in separate analyses (especially the spatial biases that seem separate from the selection biases). I wonder whether it would be possible to have the four parameters in a single analysis, or equation (a bit as we would have in a regression analysis), so that the behaviour of a given individual could be summarised in a very straightforward way (e.g. this participant’s behaviour is explained at 80% by a preference for target A and at 20% by a preference for proximity). NB: I’m not an expert in Bayesian modelling, so maybe this is something not doable, but if doable, it would be very cool.

2. It is said multiple times (l.84-87, l.306-311, and probably elsewhere), that the model can generate good parameter estimates with little trials. Is the model also resistant to the manipulation of the number of target selections per trial? Knowing this would be very useful for the study of populations such as children, who have difficulties to perform many target selections per trial (see, e.g. Olafsdottir et al., 2016, 2020).

3. I understand that the model is somehow limited by the number of target types (can only be used with two target types). More and more studies are now contrasting 3+ target types, and I would find useful to update the model so that it could be used to assess the relative preference (pA) for n = 3 or more target types.

4. In the re-analysis of the data from Kristjansson et al. (2014), the model allows a clear identification of “super-foragers” who show low pS in conjunction foraging (e.g. Fig.21 of the supplementary materials part 1). Just to be sure: are these super foragers the same individuals as the ones identified as super foragers in the original analysis? Similarly, are the participants showing a strong preference for valuable targets on supplementary figure 26 the same individuals as the ones identified in the original analysis in Tagu & Kristjansson (2022)?

5. Data used for testing the model were imported from studies involving “exhaustive” foraging (e.g. Kristjansson et al., 2014), where participants were required to select all available targets, as well as from studies involving “non-exhaustive” foraging (Dawkins, 1971; Tagu & Kristjansson, 2022), where participants only select some of the available targets. If I understand correctly, the model doesn't seem to be influenced by the differences in task instructions and can reliably generate pS and pA in both conditions. Maybe the authors would like to highlight this other strength of the model, on the top of its other qualities.

Specific comments:

- Figures: the figures often have multiple panels and which panel is presented in the text is not always very clear. Please consider labelling the panels with letters.

- l.8-9: a parenthesis seems to be missing here

- l.223-227: could the authors indicate which target type is target 0? The same colours (red, green, yellow, blue) are often used by different foraging studies. Is the preference for target type 0 a reliable preference for a given colour (e.g. red) found in several datasets, or is it specific to the dataset that is analysed here?

- l.261: typo, “figure figure 5”

- l.271-272: I must be missing something here, but the text is referring to a negative direction bias while in figure 6 I (mistakenly?) see a positive direction bias.

- Supp. Materials part 1, Figure 9: as the curves are similar, it is difficult to identify the four distributions. Maybe using line colour instead of filling colour in the legend would help identify the curves on the plot. Same thing for Fig.3 of supplementary materials part 2.

- Supp. Materials part 1: a small bias on pS is tested with a preference of .6 (end of p. 8), while a small bias on pA is tested with a preference of .8 (p. 9). But is .8 really a “small” preference? Why not using a bias of .6 in both analyses?

- Supp. Materials part 1, end of p. 17: it is said that the experiment contains 20 trials per condition, and then that the analysis was restricted to trials 5-25. So there appears to be a typo somewhere here!

**Have the authors made all data and (if applicable) computational code underlying the findings in their manuscript fully available?**

Reviewer #1: None

Reviewer #2: Yes

Reviewer #4: Yes

PLOS authors have the option to publish the peer review history of their article (what does this mean?). If published, this will include your full peer review and any attached files.

Reviewer #1: No

Reviewer #2: No

Reviewer #4: **Yes: **Jérôme Tagu

Figure Files:

Data Requirements:

Reproducibility:

References:

---

## [Editor Report · Decision Letter 1]

6 Jan 2022

Dear Dr Hughes,

We are pleased to inform you that your manuscript 'Foraging as sampling without replacement: a Bayesian statistical model for estimating biases in target selection' has been provisionally accepted for publication in PLOS Computational Biology.

Best regards,

Ulrik R. Beierholm

Associate Editor

PLOS Computational Biology

Natalia Komarova

Deputy Editor

PLOS Computational Biology

---

## [Editor Report · Acceptance letter]

20 Jan 2022

PCOMPBIOL-D-21-01844R1 

Foraging as sampling without replacement: a Bayesian statistical model for estimating biases in target selection

Dear Dr Hughes,

I am pleased to inform you that your manuscript has been formally accepted for publication in PLOS Computational Biology. Your manuscript is now with our production department and you will be notified of the publication date in due course.

With kind regards,

Livia Horvath
